# Microstructural Features in Multicore Cu–Nb Composites

**DOI:** 10.3390/ma14227033

**Published:** 2021-11-19

**Authors:** Elena N. Popova, Irina L. Deryagina, Evgeniya G. Valova-Zaharevskaya, Maria Letizia Ruello, Vladimir V. Popov

**Affiliations:** 1M.N. Miheev Institute of Metal Physics Ural Branch of RAS, 620108 Ekaterinburg, Russia; popova@imp.uran.ru (E.N.P.); deryagina@mail.ru (I.L.D.); valova@imp.uran.ru (E.G.V.-Z.); 2INSTM Research Unit, Department of Materials, Environmental Sciences and Urban Planning (SIMAU), Università Politecnica delle Marche, 60131 Ancona, Italy; m.l.ruello@staff.univpm.it; 3Institute of Metals, Technion R&D Foundation, Ltd., Haifa 3200003, Israel

**Keywords:** multifilamentary Cu–Nb composites, microstructure, microhardness, thermal stability, electron microscopy

## Abstract

The study is devoted to heavily drawn multicore Cu–18Nb composites of cylindrical and rectangular shapes. The composites were fabricated by the melt-and-deform method, namely, 600 in situ rods of Cu–18%Nb alloy were assembled in a copper shell and cold-drawn to a diameter of 15.4 mm (e = 10.2) and then rolled into a rectangular shape the size of 3 × 5.8 mm (e = 12.5). The specimens were analyzed from the viewpoints of their microstructure, microhardness, and thermal stability. The methods of SEM, TEM, X-ray analysis, and microhardness measurements were applied. It is demonstrated that, at higher strain, the fiber texture $\langle 110\rangle \mathrm{Nb}∥\;\langle 111\rangle \mathrm{Cu}∥\;\mathrm{DD}\;$(drawing direction), characteristic of this material, becomes sharper. The distortions of niobium lattice can be observed, namely, the $\left\{110\right\}{\;}_{Nb}$ interplanar distance is broadened in longitudinal direction of specimens and compacted in transverse sections. The copper matrix lattice is distorted as well, though its distortions are much less pronounced due to its recrystallization. Evolution of microstructure under annealing consists mainly in the coagulation of ribbon-like Nb filaments and in the vanishing of lattice distortions. The structural changes in Nb filaments start at 300–400 °C, then develop actively at 600 °C and cause considerable decrease of strength at 700–800 °C.

## 1. Introduction

Cu-based composites combine high strength with high conductivity, which makes them promising materials for electrical devices in which large electromagnetic and mechanical forces are exerted, particularly, for windings of high-field pulsed magnets [1,2]. In most cases, transition BCC metals with extremely low solubility in Cu (Nb, Fe, Cr, and V) or FCC Ag are used as additions to the copper matrix [3,4,5,6,7]. 

Composite wires in which copper matrix is strengthened with thin Nb filaments are of special interest among the in situ composites, and there are a lot of publications devoted to them beginning from the end of the last century up to today [8,9,10,11,12,13,14,15]. The Cu–Nb conductors demonstrate the highest values of ultimate tensile strength (UTS) up to 2000 MPa for 18–20% Nb. An effect of abnormal strengthening is observed at strains higher than 5, when the sizes of structural constituents become less than 100 nm, that is, they become commensurable with characteristic parameters of many physical quantities, such as electron free path, dislocation and disclination lengths, domain sizes in ferromagnets, etc. In addition, the density of Cu/Nb interfaces is substantially increased at high strains, and they can play a crucial role in the attainment of anomalous properties, as was well-established for different types of nanomaterials [16,17,18,19]. 

Along with high strength, Cu–Nb composites possess high conductivity (about 70% of IACS—international annealed copper standard) due to their negligible mutual solubility (<0.1 at.%) [15]. Even at 800–1000 °C, the mutual solubility is 0.15–0.20% of Nb in Cu and 0.4–0.5% Cu in Nb. That is why they are promising for application in electromagnets with high (up to 100 T) fields, NMR devices with fields of 15–20 T, in electrical supply networks subjected to extremal mechanical forces, in aviation and space technologies, etc. Additionally, Cu–Nb composites are of special interest as reinforcing material for Nb_3_Sn-based superconductors [20,21,22,23]. 

For more than two decades, various types of winding materials based on Cu–Nb microcomposites were worked out at the Bochvar Institute of Inorganic Materials (Moscow, Russia) [10,11,12,13]. At first, they were fabricated in form of thin single-core in-situ wires. Specific features of their texture, microstructure, and possible mechanisms of strengthening were studied and analyzed in a number of our earlier publications [24,25,26,27]. Later a multicore geometry was applied for fabrication of high-strength Cu–Nb wires with various shapes of transverse sections, the sizes of up to several millimeters [13]. Specimens of such wires with different true strains have been studied in the present research. 

According to [13], the multicore Cu–18Nb composite does not change at room temperature for several years, and its structure and properties do not degrade at heating to 200 °C. In real operation conditions, the wires may be heated up to much higher temperatures, which can result in the loss of their strength. Additionally, manufacturing of in situ composites is a multistage process, and intermediate anneals are required between drawing stages in order to avoid fracturing. Therefore, it is necessary to know the permissible limit of the annealing temperature. Prolonged diffusion annealing at temperatures up to 675 °C is used in fabrication of multifilamentary Nb_3_Sn-based superconductors, in which, as mentioned above, Cu–Nb composites can be used as reinforcing inserts. That is why the problem of thermal stability of Cu–Nb composites is of great importance. The loss of their strength at annealing is mostly associated with modification of the specific structure of Nb filaments in the copper matrix, namely, with transformation of ribbon-like filaments into bamboo-like ones, that is, their spheroidizing and coarsening [28,29,30]. In this regard, the microstructural investigation of composite at different annealing temperatures, which allows specifying the temperature of the beginning and end of softening after different true strains before the annealing, is of significant interest.

The main goals of this research are to reveal the microstructural features of multicore Cu–18Nb composites and the state of Cu/Nb interfaces dependently of their true strains, to demonstrate the evolution of texture under cold deformation and subsequent annealing, and to estimate the thermal stability of the structure from the view-point of structural changes under annealing accompanied with the loss of strength. 

## 2. Materials and Methods

Samples of multicore Cu–18Nb microcomposites were developed and manufactured at JSC VNIINM by the melt-and-deform method [13]. Ingots of the initial Cu–18%Nb alloy were vacuum-melted with high-purity electron-beam melted copper (99.99%) and electron-beam melted niobium (99.9% Nb). The ingots were extruded into rods and drawn into hexahedrons the size of 5.4 mm. Then, the rods were cut into pieces, and 600 pieces of rods were assembled in a copper shell and cold-drawn to a diameter of 15.4 mm. This cylindrical-shaped composite is denoted as Sample 1, and its true strain e is 10.2. Then, it was rolled into a rectangular shape the size of 3 × 5.8 mm^2^, which gave Sample 2 with e = 12.5. The UTS (ultimate tensile strength) and YS (yield strength) of the rectangular-shaped composite (Sample 2) are, respectively, 1100 and 720 MPa and practically do not decrease up to 200 °C, and its resistance ratio between 293 and 77 K (R293/R77) is 4.5. The samples were annealed for 1 h in the temperature range 300–800 °C in a vacuum to examine the thermal stability of the microstructure. 

The composites microstructure and composition before and after annealing was studied by scanning (SEM) electron microscopy (on Quanta-200 and Inspect F microscopes, FEI Company, Hillsboro, USA) with an EDAX microanalysis attachment, in transverse and longitudinal sections. Additionally, transmission electron microscopy (TEM) (on JEM-200 CX microscope, JEOL, Tokyo, Japan) was applied on longitudinal and transverse foils of the samples. To make polished sections for electron-microscopic studies, thin plates were cut from the samples on an electric-spark machine, then ground on abrasive materials with decreasing grain size and mirror-polished. For TEM studies, mechanically thinned foils were polished in a mixture of acids (3 HNO_3_:2 H_2_SO_4_:1 HF).

X-ray data were obtained in a DRON diffractometer, in Cr (K_α1_ + K_α2_) radiation, in the range of angles 25 ≤ 2 Θ ≤ 140 degrees, in longitudinal and transverse sections of samples before and after annealing in a vacuum at 800 °C for 1 h, and on Nb filaments etched out of the Cu matrix. A pure Nb rod and Cu powder obtained by electrolytic precipitation served as the reference standards. 

Microhardness was measured in transverse polished sections of composite specimens using a specialized unit of an optical microscope Neophot-21 and calculated as $H\;=18192P/L^{2}$ MPa, where P stands for the load in grams, and L stands for the indentation diagonal in μm. An error of an indentation diagonal measurement was calculated as a random component of multiple determinations with a reliability level of 0.95. The relative error of microhardness measurements was calculated as an error of indirect determinations and comprised 3–5%.

## 3. Results

Transverse sections of Samples 1 and 2 of a multicore Cu–18Nb composite (SEM images taken with different magnifications) are shown in Figure 1. The following features can be noted. Under the multistage drawing and assembling, the deformation is distributed non-uniformly over the cross-sections. In the cylindrical Sample 1, the hexagonal strands are more distorted closer to the periphery (Figure 1a), and in the rectangular Sample 2 along the diagonals (Figure 1d). In addition, within each strand, alternating lighter and darker rings are visible in both samples (Figure 1b,e). These rings indicate non-uniform distribution of Nb filaments in the Cu matrix throughout the transverse sections of strands. According to microanalysis, in the lighter circular zones, there are more Nb filaments with smaller spacing between them than in darker zones. The main feature is the complex morphology of curved niobium ribbon-shaped filaments (Figure 1c,f). This morphology has been observed in various types of Cu–Nb composites and is attributed to the peculiarities of slipping systems in the BCC Nb and the influence of the FCC copper matrix [3,8,11,31,32,33,34,35].

In Sample 1, the thickness of the Nb ribbon-like filaments ranges from 40 to 150 nm with an average value of 70 nm, whereas the distance between the ribbons varies over a very wide range, from hundredths of a micron to 1 μm. An increase in true strain to 12.5 results in an increase of the Nb-ribbons’ density in the copper matrix, and their average thickness reduces to 30 nm. The spacing between ribbons in the regions with the lowest density does not exceed 200 nm. 

As the niobium ribbons become thinner and the distances between them become shorter under higher strain, the area of Cu/Nb interfaces increases, which, as shown in a number of publications (see, for example, [2,3,4,11,33,34]), causes an increase in microhardness and ultimate strength. Indeed, the microhardness increases from 2400 MPa in Sample 1 (*e* = 10.2) to 3300 MPa in Sample 2 (*e* = 12.5).

The SEM data on microstructure of composites under study are confirmed and complimented by the results of TEM investigations (Figure 2 and Figure 3). The Nb ribbons in Sample 1 are thicker than in Sample 2, their thickness being 70–80 and 30–40 nm, respectively. In the cross-sections, the Nb ribbons have an intricate curved shape (Figure 2a and Figure 3a); they bend around the grains of the copper matrix, which in both samples have a polyhedral shape, the sizes of 200–300 nm, and low dislocation density (Figure 2b and Figure 3c). Such structure of the composite matrix can be explained by the dynamic recrystallization of copper. In some SAEDs (selected area electron diffraction patterns), the reflections of Cu and Nb are located in the corresponding Debye rings (Figure 3b), and on the others, one of the planes of the reciprocal lattice of Cu can be distinguished (Figure 2c).

In some SAEDs, the (110)_Nb_ reflections form a diffuse ring (Figure 3b), indicating the presence of amorphous areas at Nb/Cu interfaces, as was also observed by other authors [36]. In addition, we note that, when calculating a large number of SAEDs of both samples from both longitudinal and cross-sections, it was found that the (110)_Nb_ interplanar distances (with tabular value of 2.33 nm) vary within a fairly wide range of 2.30–2.40 nm. The accuracy of calculating electron diffraction patterns is surely not high, but, nevertheless, it can be concluded that in some areas there are distortions of the niobium lattice, and it is either stretched or compressed. The results of a much more accurate X-ray analysis presented below confirm this conclusion.

In lengthwise sections of both samples, alternating layers of Cu and Nb are seen (Figure 3d,e), the dislocation density in Nb filaments being significantly higher than in the Cu. The $\langle 110\rangle \mathrm{Nb}$ and $\langle 111\rangle \mathrm{Cu}$ directions are in line with the drawing direction, with a misorientation of 2–5° relative to each other (Figure 3f). The same orientation relationships between these directions were found in Cu–Nb composites obtained by bundle-and-deform and melt-and-deform (in situ) methods in [37,38]. If, under wire drawing, the Cu matrix and Nb filaments were independently rearranged so that their close-packed planes, $\left(111\right)\mathrm{Cu}$ and $\left(110\right)\mathrm{Nb}$, were perpendicular to the drawing direction, then there would be no misorientation between them. Therefore, the authors of [37] concluded that there was a certain orientation relationship between the filaments and matrix and that the Nb/Cu interfaces became partially coherent in the process of large plastic deformation. 

According to the data of X-ray analysis, the pronounced fiber texture ${\langle 110\rangle }_{\mathrm{Nb}}∥{\langle 111\rangle }_{\mathrm{Cu}}∥\mathrm{DD}$ is established in both samples, which is in agreement with numerous publications [24,25,26,38,39,40,41,42]. There is also an additional texture in copper, ${\langle 200\rangle }_{\mathrm{Cu}}∥\mathrm{DD}$. It is weaker, and many authors did not notice it, though, in a number of studies, it was observed as well [24,25,40,41]. The texture of both composite constituents was determined from the weight (P_hkl_) of their main X-ray peaks, the formula for which is given in [31]. With strain increasing from 10.2 to 12.5 (Samples 1 and 2), the weight of (111)_Cu_ peak increases from 1.8 to 2.0, and that of (110)_Nb_ from 1.9 to 2.1, which indicates synchronous texture increase in copper matrix and niobium filaments (Table 1).

The development of microstresses in copper and niobium with the strain increasing were estimated from the FWHM (full width at half maximum) parameter. The FWHM of the (111)_Cu_ peak in Samples 1 and 2 in comparison with the reference specimen increases by 1.4 and 1.9 times, respectively, and that of the (110)_Nb_ peak increases by 2.2 and 2.8 times. Thus, the microstresses are present in both constituents of the composite, and they increase with increasing strain. In Nb filaments, the level of these stresses is higher than in copper.

Along with the noticeable broadening of X-ray lines, corresponding to the presence of internal stresses in both constituents of the composite, a shift of the peaks relative to the reference values, especially in niobium, was also found (Figure 4). The peak shifts indicate lattice distortions. The shift of the ${\left(110\right)}_{\mathrm{Nb}}$ peak depends both on the section plane (transverse or longitudinal) and on the strain degree. In the diffraction patterns from the cross-sections, the (110)_Nb_ peak is shifted towards the smaller, and from the longitudinal sections, towards the larger Bragg angles (Figure 4a), and with the strain increasing, these shifts become larger.

The Nb crystal lattice distortion is maximal in the $\langle 110\rangle {\;}_{\mathrm{Nb}}$ direction. The Nb interplanar d_110_ distances are elongated in the drawing direction and compressed in the transverse areas. The distortion (Δ*d*) in the <110> direction was attributed to the slight change in the interplanar spaces $d_{110}\left(Nb\right)$ in Samples 1 and 2 in comparison with the reference value $d_{110}\left(Nb^{*}\right)$ and was calculated as: $$\Delta d=\frac{d_{110}(Nb*)-d_{110}(Nb)}{d_{110}(Nb*)}•100\%$$

With the true strain increasing from 10.2 to 12.5, the ${\left(110\right)}_{\mathrm{Nb}}\;$ interplanar space distortions both in longitudinal and transverse sections increase by more than a factor of 3 (Table 1).

As for the distortions of the Cu lattice, they are much less pronounced (only several tenths of a percent). In this case, two features should be noted. First, as in niobium, the distortions increase with an increase in the true strain, and this is quite predictable. However, in contrast to niobium, there is no regularity in the tension–compression of the copper lattice, depending on the section direction (longitudinal or transverse). It can be assumed that this is due to several factors. Firstly, this is the absence of a pronounced ribbon shape, contrary to that of niobium. Secondly, as noted above, the second component of the texture is also possible in copper, which can weaken with an increase in the degree of deformation [24,25,40,41]. In general, since the distortions in copper lattice are so small, it can be assumed that only small boundary regions are distorted, and the lattice is not distorted in the entire grain body, and these boundary regions can be distorted in different ways relative to the deformation axis. 

It should be noted that the results obtained on the lattice distortions of niobium and copper in such composites agree with the results of other authors [11,14,42,43]. Thus, for example, according to [42], in the Cu–18Nb composite in the range of true strains 8.8 > e > 9.6, pronounced distortions of the niobium (110)_Nb_ lattice appear (compression in the direction perpendicular to the drawing axis and tension parallel to this axis), leading to an enhancement in the degree of mismatch between the ${\left(110\right)}_{\mathrm{Nb}}$ and ${\left(111\right)}_{\mathrm{Cu}}$ lattices with an enhance in the strain degree. Here, the strength of the composite increases, and regions of coherent conjunction of the copper and niobium lattices appear. In [14], lattice distortions were found in both niobium and copper, and the authors of this work believe that, along with a large area of Cu/Nb interfaces, these distortions make an additional contribution to strengthening. In [43], large distortions of the lattices of both copper and niobium were found in a Cu–Nb multilayer system obtained by magnetron sputtering.

The behavior of Cu–18Nb composites (Samples 1 and 2) under heat treatment was studied, after which their microhardness and microstructure were investigated. Figure 5 illustrates the microhardness behavior under annealing. Over the entire temperature range, the composite with higher strain exhibits higher microhardness but sharper decrease in it. A similar behavior was found in niobium processed by high pressure torsion, namely, after the higher degree of deformation, the sharper drop in microhardness was observed in the annealing range of 400–600 °C [44]. 

In the range from room temperature to 300 °C, the microhardness of both samples decreases insignificantly. More pronounced decreasing of microhardness is observed in the temperature range from 300 to 500 °C. In the range of 600–700 °C, the microhardness decrease becomes sharper, and at 700 °C, the microhardness reaches its minimum value, remaining approximately at the same level at an annealing temperature of 800 °C. Thus, the softening starts at 300 °C and practically ends at 700 °C. It is interesting to note that, in copper and niobium processed by severe plastic deformation [45,46,47], as well as in the Cu–18Nb composite obtained by the bundle-and-deform method [48], the microhardness demonstrates similar behavior under annealing. However, taking into account the relative decreasing of microhardness, it can be concluded that the composites studied in this work are thermally more stable than copper and niobium processed by SPD [46,47]. On the other hand, they are less stable than Cu–15Nb with the true strain of 6.0 [28] and Cu–18Nb with the true strain of 3.5 [48].

The texture of Nb and Cu observed before annealing is retained after the annealing in all the temperature range studied, although the degree of niobium texture under the annealing slightly decreases, which agrees, for example, with the conclusions made in [28]. At the same time, the positions of the Nb peaks and their half-widths in the X-ray spectrum of Samples 1 and 2 after the annealing at 800 °C considerably change in comparison with the deformed samples, namely, the shifts of the (110)Nb line disappear and the half-widths of the Nb peaks decrease (approximately by a factor of 1.5). All this indicates vanishing of macrostresses and weakening of microstresses in Nb. After the annealing at 800 °C, the Nb interplanar spaces completely coincide with the reference ones.

The above-described changes in the properties of the composites under study are caused by the change in the microstructure of ribbon-like Nb filaments. According to SEM data, under the annealing at 800 °C, coagulation of the Nb ribbons occurs, which leads to broadening and rounding of ribbons in the cross-sections and the establishment of numerous constrictions and breaks in the lengthwise sections of the specimen (Figure 6). Several measurements indicated as an example in Figure 6a demonstrate that the diameters of round-shaped filaments range from about 150 to 400 nm, and in the filaments with still elongated shape, the aspect ratio of their width to thickness is only about 2.5–3.5, whereas, in the deformed state, it can reach more than 30 [28]. Thus, after the annealing, the so-called bamboo-like structure of niobium ribbons is formed. Such modification in the microstructure of niobium ribbons leads to the destruction of their bonds with the copper matrix in the region of the Nb/Cu interfaces, which results in the decrease in the strength of the composites. 

The results of TEM studies are shown in Figure 7, Figure 8 and Figure 9. Some thickening of the niobium ribbons in the cross-sections of Sample 2 is seen after the annealing at 300 °C (Figure 7a). Further development of this process is observed after the 400 °C annealing (Figure 7b). The Nb filaments are not uniform in thickness, their edges being darker and middle parts lighter. Thus, the edges are less transparent to the electron beam of the microscope, which indicates their distortions and the loss of coherence at Nb/Cu interfaces. At 500 °C, the ribbons of niobium become rounded in cross-sections (Figure 7c), which is evidence of the beginning of the bamboo-like structure formation (niobium ribbons transform into cylinders).

When the annealing temperature is increased, the process of coagulation develops, capturing an increasing number of Nb ribbons (Figure 8). After the annealing at 600 °C, all ribbons have a considerably thickened or rounded shape (Figure 8a). In longitudinal sections of this composite, the formation of the bamboo-like structure is observed (Figure 8b,c). Here, Nb-ribbons are mainly free of dislocations, which remain only in the places of constrictions.

When the annealing temperature is increased to 700–800 °C, the coagulation of the Nb filaments is completed, and their transverse dimensions increase up to 120 nm (Figure 9a,b). In some areas, the Nb ribbons demonstrate the bamboo-like structure, with larger diameters of cylinders and narrower necks (Figure 9c), and in other areas, the ribbons break, and separate cylindrical fragments are formed. The electron diffraction patterns from the cross-sections contain reflections corresponding to reciprocal lattices of both copper and niobium. In the longitudinal sections, there are regions where niobium ribbons are etched from the copper matrix, and in SAEDs, the reflections correspond only to niobium. These data indicate the destruction of coherent bonds at Nb/Cu interfaces under the annealing temperatures of 700–800 °C, which explains the sharp decrease in the microhardness of the composite after annealing at 700 °C.

The experimental findings and microstructural results on the appearance and development of niobium filaments coagulation and the microhardness decreasing with an enhance in the annealing temperature of Cu–Nb multi-core composites agree with numerous publications [13,28,29]. The present study allows us to clarify the onset of spheroidization of niobium filaments (500 °C), which turns out to be lower than, for example, in [28,29], in which the change in the morphology of niobium filaments was estimated only from SEM data. Thus, according to these publications, coagulation of Nb filaments in Cu–15Nb and Cu–18Nb composites was observed after the 700 °C annealing, although the microhardness decreasing was noted at significantly lower temperatures (200 °C according to [28] or 450 °C according to [29]). The microhardness decreasing indicate that modification of the Nb ribbons occur at lower temperatures, just as is observed in the present paper.

## 4. Conclusions

The structure and texture of in situ multicore Cu–18Nb composites have been studied after their deformation with true strains of 10.2 and 12.5 and subsequent annealing. The main conclusions are as follows.

(1)The axial texture $\langle 110\rangle \mathrm{Nb}\;∥\langle 111\rangle \mathrm{Cu}\;∥\;\mathrm{DD}\;$(drawing direction) is formed under the deformation and retained, although it is weakened, in all the annealing temperature ranges studied. (2)Microstresses and macrostresses in ribbon-like Nb filaments increase with the true strain increasing. The macrostresses result in distortions of the Nb lattice-stretching of (110)Nb interplanar distances along the drawing axis and their compression in the direction perpendicular to it, and these distortions noticeably increase with an increase in strain.(3)In the copper matrix, a high level of microstresses has also been found, which increases with the true strain increasing. Small macrostresses are also present, which lead to distortions of the Cu lattice, although they are an order of magnitude lower than in Nb.(4)Under the annealing, coagulation of Nb filaments (their thickening and rounding in cross-sections) occurs. This process begins at 300–400 °C, develops actively at 600 °C, and practically completes at 700 °C. The necking and rupture of Nb filaments under annealing results in the decrease of microhardness. (5)The thermal stability of Cu–18Nb microcomposites is higher than that of their constituents (Cu and Nb) processed by high-pressure torsion, which may be due to the coherency of Cu/Nb interfaces.

## Figures and Tables

**Figure 1 materials-14-07033-f001:**
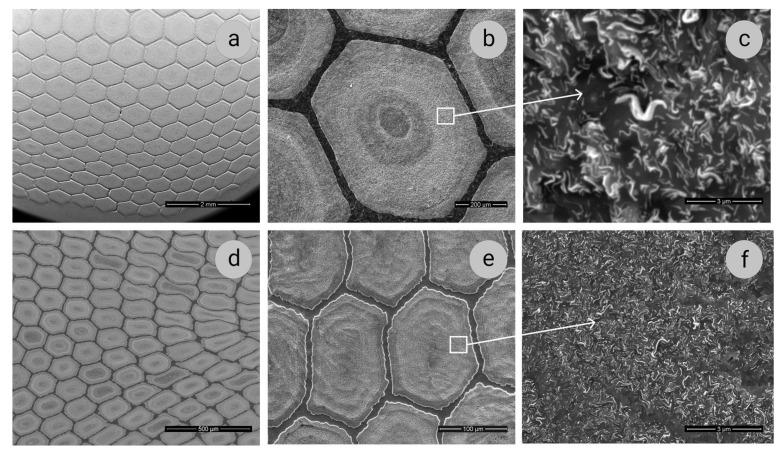
Transverse sections of Samples 1 (**a**–**c**) and 2 (**d**–**f**) of multicore Cu–18Nb composite (SEI images). The areas taken with higher magnification (Figure 1c,f) are denoted with squares in Figure 1b,e.

**Figure 2 materials-14-07033-f002:**
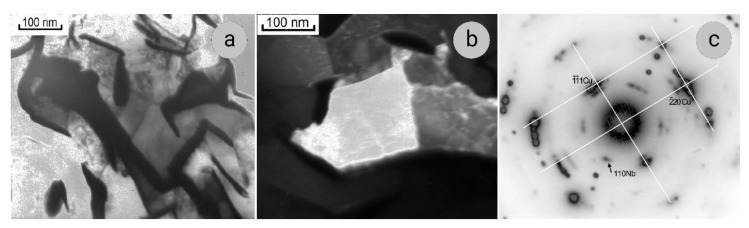
Microstructure of transverse section of Cu–18Nb composite, Sample 1: (**a**)—bright-field image; (**b**)—dark-field image in (220)_Cu_ reflection; (**c**)—SAED, zone axis [112]_Cu_.

**Figure 3 materials-14-07033-f003:**
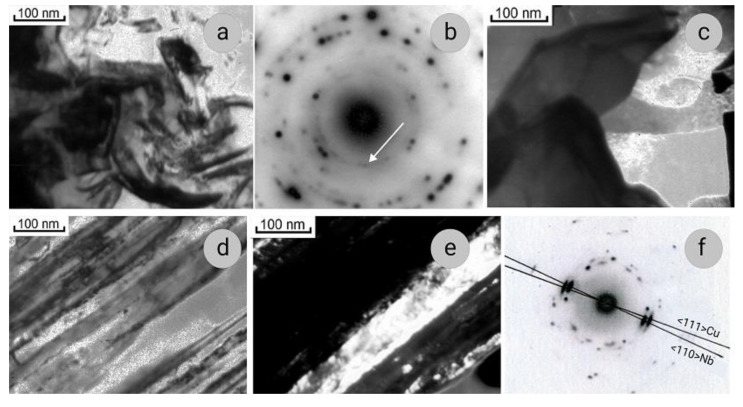
Microstructure of transverse (**a**–**c**) and longitudinal (**d**–**f**) sections of Cu–18Nb composite, Sample 2: (**a**,**d**)—bright-field images; (**b**)—SAED; (**c**)—dark-field image in (220)Cu reflection; (**e**)—dark-field image in (110)Nb and (111)Cu reflections; f- SAED, zone axes [110]Nb and [111]Cu.

**Figure 4 materials-14-07033-f004:**
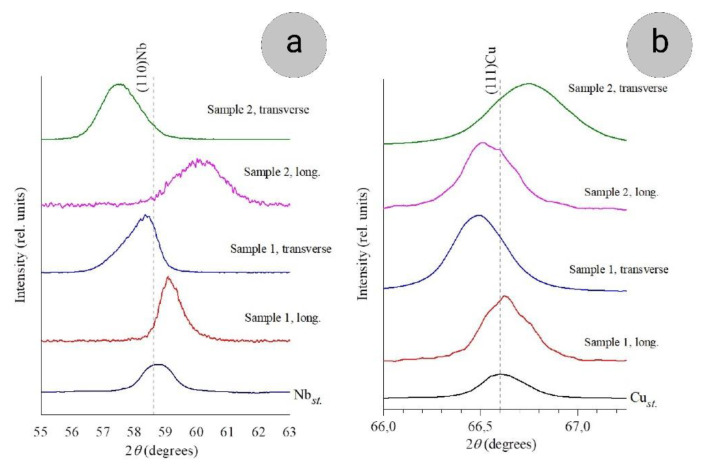
Positions of (110)_Nb_ (**a**) and (111)_Cu_ (**b**) peaks in composites and standards.

**Figure 5 materials-14-07033-f005:**
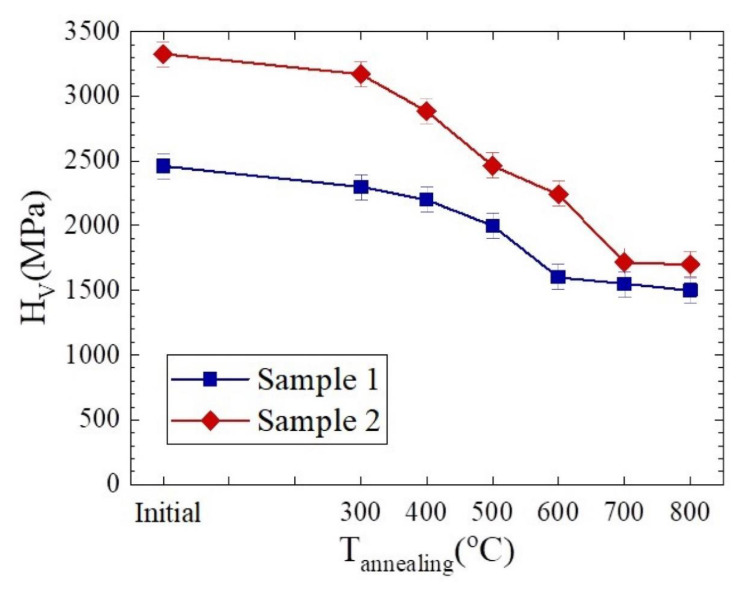
Microhardness of Samples 1 (■) and 2 (♦) of Cu–18Nb composites versus the annealing temperature.

**Figure 6 materials-14-07033-f006:**
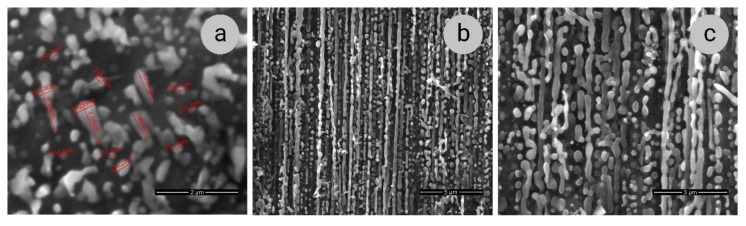
SEM images of Sample 2 (*e* = 12.5) after the annealing at 800 °C, 1 h: (**a**)—transverse section; (**b**,**c**)—longitudinal sections.

**Figure 7 materials-14-07033-f007:**
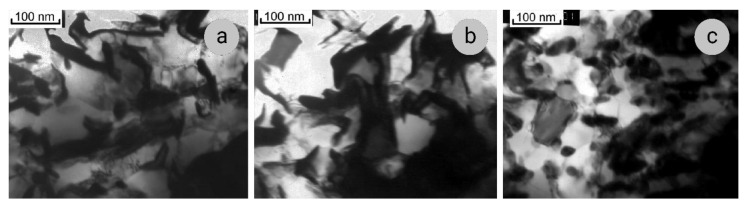
The structure of Sample 2 (*e* = 12.5) in transverse sections after the annealing for 1 h at 300 °C (**a**), 400 °C (**b**), and 500 °C (**c**).

**Figure 8 materials-14-07033-f008:**
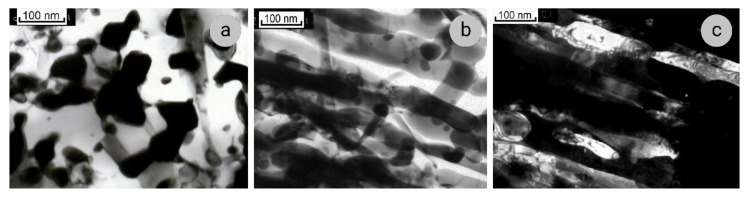
TEM images of Sample 2 (*e* = 12.5) annealed at 600 °C, 1 h: (**a**)—transverse section; (**b**,**c**)—longitudinal sections; c dark-field image in (110)_Nb_ and (111)_Cu_ reflections.

**Figure 9 materials-14-07033-f009:**
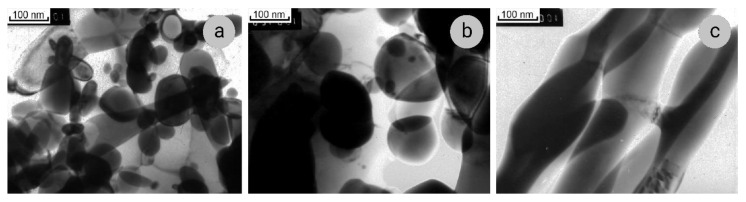
The structure of Sample 2 (*e* = 12.5) after annealing at 700 °C, 1 h (**a**) and 800 °C, 1 h (**b**,**c**): (**a**,**b**)—transverse sections, (**c**)—longitudinal section.

**Table 1 materials-14-07033-t001:** Characteristics of samples based on the X-ray data.

Sample	*e*	Wire Section	P_111Cu_, %	P_110Nb_, %	FWHM (111)_Cu_,°	FWHM (110)_Nb_,°	d_110Nb_ Ǻ	Δd_110Nb_, %	d_111Cu_ Ǻ	Δd_111Cu_, %
1	10.2	transverse	84.0	91	0.34	1.2	2.350	−0.6	2.089	−0.15
1	10.2	longitudinal					2.320	0.6	2.085	0.04
2	12.5	transverse	94.0	99	0.46	1.5	2.381	−2.0	2.082	0.19
2	12.5	longitudinal					2.289	2.0	2.088	−0.10
Nb_st_	-	-			-	0.5	2.335		-	
Cu_st_	-	-			0.24	-	-		2.086	

## Data Availability

The raw/processed data required to reproduce these findings cannot be shared at this time due to technical or time limitations.

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
