# Peer review of "Microstructural Features in Multicore Cu–Nb Composites"

_materials, 2021, doi:10.3390/ma14227033_

Round 1

Reviewer 1 Report

This  paper is a good work for Cu- Nb f filaments composites.

But the authors manufactured this composite for electro magnetic applications, so some characterizations are mandatory  to make a  full and satesfied  work in this fieh, those are

1- Measuring the electrical conductivity of sintered samples

2- Estimating the electromagnetic properties by using the VSM  { Viberating sample magnetometer} device  ti study the effect of Nb  fiaments on the magnetism of the prepared composites

3- Also, as the authors  have estimated the thermal stability, so thermal conductivity must be evaluated

The experimental work must be contains more details about the manufacturing process.

 The abstract must be repeated for more details about the experimental work

Author Response

We would like to express our deepest gratitude to the Reviewer for reviewing our manuscript and providing useful comments for its improvement. We also thank you for your expert opinion and your valuable comments. Please, find attached our response point-by-point to your comments.

Reviewer 2 Report

1. For Fig. 1c and f, it is better having the same magnification if they are compared.
2. why chose the true strain e=10.2 and 12.5 for these two types of samples? What is relationship with the microstructure?

3.  Under the similar true strain, why the ribbon size and thickness are big difference?

4. Fig. 2 a and b are the same region taken for TEM? If not, it should use the BF and DF TEM taken from the same region.

Author Response

We want to thank the Reviewer for his valuable time and professional comments. We have performed the thoroughful modification of the manuscript and answered point-by-point to the Reviewers comments. Please find attached the response to reviewer. 

Reviewer 3 Report

The paper entitled “Microstructural features in multicore Cu-Nb composites” focuses on the microstructural features and thermal stability of multicore Cu-18Nb specimens. The samples were subjected to annealing for 1 h in the temperature range 300-800°C in a vacuum to examine the thermal stability of the microstructure. Characterization techniques such as SEM coupled with EDS, XRD, TEM, microhardness measurements, have been conducted. The paper may be interesting from a scientific and practical point of view.

I would like to recommend the publication of the manuscript in this journal after fulfilling the following recommendations:

  1. The aim of the study should be better formulated.
  2. More information about the shape, organization, size and production of Cu-18Nb multicore microcomposite namely sample 1 and 2, should be given in the “Materials and Methods” section. Except for the shape and true strain, what is the difference between them?
  3. The preparation of the samples for TEM measurements is not given in the text.
  4. In fig. 1, it should be mentioned if BEI or SEI images are used. Additionally, it is not clear where is the position of these “ribbon-like filaments” within the examined samples? Probably, an initial image at lower magnification should be given.
  5. The microhardness results seem statistically unreliable. The average hardness values should be presented together with their standard deviation values.
  6. The measurements, indicated in Fig. 6 a, are not mentioned either in the text ot figure caption.
  7. The solid-state solubility between Cu and Nb should be at least mentioned in the text.
  8. The conclusions should be re-formulated. Instead of a conclusion, there is a summary of the results.
  9. In some places, the English style and grammar have to be improved. Some examples of inadequate style: “According to microanalysis, the lighter rings are more densely populated with the Nb filaments”; “Then, with the annealing temperature increasing from 300°C to 500°C, the slope of the microhardness curve increases.…..”; “……the microhardness in general decreases in the same manner with an increasing annealing temperature.” …and some more….

Author Response

(The authors gave the same response as above.)

Round 2

Reviewer 2 Report

Authors addressed the comments.

Reviewer 3 Report

The authors have carefully addressed all the review’s recommendations and the manuscript has been substantially improved.